Corrected: Author correction

# Ultrafast switch-on dynamics of frequency-tuneable semiconductor lasers

Iman Kundu [1], Feihu Wang[2], Xiaoqiong Qi[3], Hanond Nong[2], Paul Dean[1], Joshua R. Freeman [1], Alexander Valavanis [1], Gary Agnew[3], Andrew T. Grier [1], Thomas Taimre[4], Lianhe Li [1], Dragan Indjin[1], Juliette Mangeney[2], Jérôme Tignon[2], Sukhdeep S. Dhillon[2], Aleksandar D. Rakić [3], John E. Cunningham [1], Edmund H. Linfield [1] & A. Giles Davies [1]

Single-mode frequency-tuneable semiconductor lasers based on monolithic integration of multiple cavity sections are important components, widely used in optical communications, photonic integrated circuits and other optical technologies. To date, investigations of the ultrafast switching processes in such lasers, essential to reduce frequency cross-talk, have been restricted to the observation of intensity switching over nanosecond-timescales. Here, we report coherent measurements of the ultrafast switch-on dynamics, mode competition and frequency selection in a monolithic frequency-tuneable laser using coherent time-domain sampling of the laser emission. This approach allows us to observe hopping between lasing modes on picosecond-timescales and the temporal evolution of transient multi-mode emission into steady-state single mode emission. The underlying physics is explained through a full multi-mode, temperature-dependent carrier and photon transport model. Our results show that the fundamental limit on the timescales of frequency-switching between competing modes varies with the underlying Vernier alignment of the laser cavity.

---

[1] School of Electronic and Electrical Engineering, University of Leeds, Leeds LS2 9JT, UK. [2] Laboratoire Pierre Aigrain, Département de physique de l'ENS, École normale supérieure, PSL Research University, Université Paris Diderot, Sorbonne Paris Cité, Sorbonne Universités, UPMC Univ. Paris 06, CNRS, 75005 Paris, France. [3] School of Information Technology and Electrical Engineering, The University of Queensland, Brisbane, QLD 4072, Australia. [4] School of Mathematics and Physics, The University of Queensland, Brisbane, QLD 4072, Australia. Correspondence and requests for materials should be addressed to I.K. (email: I.Kundu@leeds.ac.uk)

Single-mode frequency-tuneable semiconductor lasers are an important technology that have found applications across the electromagnetic spectrum. Single mode lasing operation is commonly realised through the use of a distributed feedback photonic grating, uniformly patterned along the laser cavity[1]. However, the range of frequency tuning that can be achieved in distributed feedback lasers is restricted since the grating period is fixed and there is only a limited degree by which the refractive index of the underlying laser gain material can be varied. An alternative approach that has been used to demonstrate wideband frequency tuning from a monolithic source, and without the need for external optical components[2–6], is based on a Vernier selection scheme. These lasers typically comprise a number of coupled cavity sections that each support a Fabry–Pérot comb of frequencies. Laser emission is favoured at the frequencies for which the longitudinal modes from each comb coincide, owing to a reduction both in the mirror losses and the lasing threshold at the coincident frequencies[7]. The laser frequency can furthermore be tuned over a wide range through applying a small perturbation to the refractive index of one or more of the cavity sections. These laser devices may also incorporate splitters[8], ring resonators[9] or micro-disks[10], as well as chirped photonic gratings such as sampled-gratings[11], super-structure gratings[12] or digital super-mode gratings[13].

Although the steady-state frequency tuning characteristics of such tuneable lasers are well reported, there has been no experimental investigation into the ultrafast switch-on dynamics, mode competition and frequency selection dynamics in any frequency-tuneable semiconductor laser. This restricts our understanding of such lasers and their inherent stabilisation and switching times, as well as our understanding of fundamental processes such as frequency cross-talk and non-linear optical effects in these devices. A theoretical study of the temporal variation of the optical power distribution among multiple cavity modes, including switching dynamics, in a frequency-tuneable coupled-cavity laser, and their effect on the side-mode suppression ratio was reported in ref.[14]. However, due to the lack of suitable ultrafast detection schemes, experimental investigations have been restricted to the observation of switching on timescales longer than the ~300 ps investigated theoretically in ref.[14]; for example in wavelength-division multiplexing systems where mode switching was measured over the timescale of 3−8 ns[15].

In this paper we report measurements of the temporal evolution, over picosecond and nanosecond timescales, of transient multi-mode emission into steady-state single-mode emission in a monolithic frequency-tuneable quantum cascade laser (QCL) operating at ~2.8 THz[6,16]. This is achieved by exploiting a technique for phase-resolved sampling of the terahertz field emission on sub-picosecond timescales[17], which has previously been applied to measurement of gain recovery times[18] and modelocked pulse widths[19] in terahertz lasers. The observations reported here are likely to be valid for any frequency-tuneable laser based on the Vernier effect, and are in agreement with established simulation models, for example, those reported in refs. [14,20]. Additionally, our measurements reveal a systematic variation of the laser switch-on time as a function of the comb alignment and mirror losses.

## Results

**Steady-state intensity measurements.** For this work we adopted a simple coupled-cavity design, comprising two cavity sections that are coupled optically through a narrow air gap. In this scheme, each cavity section supports a frequency comb of longitudinal modes. One section is electrically driven above threshold and forms the active lasing cavity, whereas the other section is electrically driven sub-threshold to form a passive tuning cavity. The air gap and the passive tuning cavity can be modelled together as an effective mirror with frequency-dependent reflectivity and loss parameters. This way the coupled-cavity laser can be described as a single cavity with an effective complex mirror. Lasing is favoured at discrete frequencies both supported by the lasing cavity and for which the effective mirror losses are a minimum; this condition is equivalent to modes of the respective frequency combs aligning in frequency. To tune the emission frequency, the refractive index of the tuning cavity is adjusted through controlled localised Joule heating. This is achieved by driving the tuning cavity with wide current pulses, and by varying the amplitude of the current pulses. This changes the frequency-dependence of the effective mirror losses, or equivalently changes the free spectral range of the longitudinal modes supported by the tuning cavity, resulting in tuning of the emission. The emission frequencies, spectral coverage and frequency tuning range realised from such coupled-cavity lasers is optimised through careful selection of the geometry such as the lengths and the ratio of the cavity lengths[6,16] as well as the tilt of the coupled facets[4].

The device was modelled, in the first instance, using a transfer matrix model based on scattering matrices to simulate the eigenfrequencies in the coupled-cavity laser. A dynamic reduced rate equation model[20] was used to simulate the spectral power distribution among the different eigenfrequencies, with the carrier lifetimes having been obtained from an energy-balanced Schrödinger–Poisson scattering transport calculation. The model includes the interaction between photon density and electron population for different subbands in the QCL at multiple Fabry–Pérot modes, and includes thermal effects. As such, it can simulate not only the steady-state tuning characteristics, but also temporal dynamics of mode selection. A detailed description of the numerical models is presented in the Methods section.

A coupled-cavity QCL with a 1.38-mm-long active lasing cavity, a 3.43-mm-long passive tuning cavity and a 13-μm-long air gap was fabricated, with further details presented in the Methods section. This geometry was designed such that mode hopping between two emission frequencies could be controlled through a large change in the current supplied to the tuning cavity. Although this selection of cavity lengths resulted in a relatively small frequency tuning range (~60 GHz), fine control of the alignment between the frequency combs was possible, which was exploited to probe the dynamics of mode selection.

The free-running steady-state emission spectra were measured using Fourier-transform infrared spectroscopy with bolometric detection (see Methods). A mode hop from 2.825 to 2.765 THz was observed as the current supplied to the tuning cavity was increased in the range 0–2 A (Fig. 1a). The power at each lasing mode and the side mode suppression of the output spectra are plotted in Fig. 1b as a function of tuning current. A side mode suppression ratio in the range 35–40 dB was observed across the range of tuning currents, except close to the current at which the mode hop occurs, for which the emission is multi-mode. The experimentally observed emission spectra agree well with simulations (Fig. 1c).

**Measurement of ultrafast dynamics with a cold tuning cavity.** The temporal dynamics of the laser emission were characterised using an injection seeding technique[17] permitting coherent, phase-resolved measurements of the terahertz electric field. A schematic illustration of the experimental arrangement is shown in Fig. 2. The phase-stable seeding pulse, generated by illuminating a broadband photoconductive antenna with a femtosecond laser, was injected into the lasing cavity. To ensure that the laser

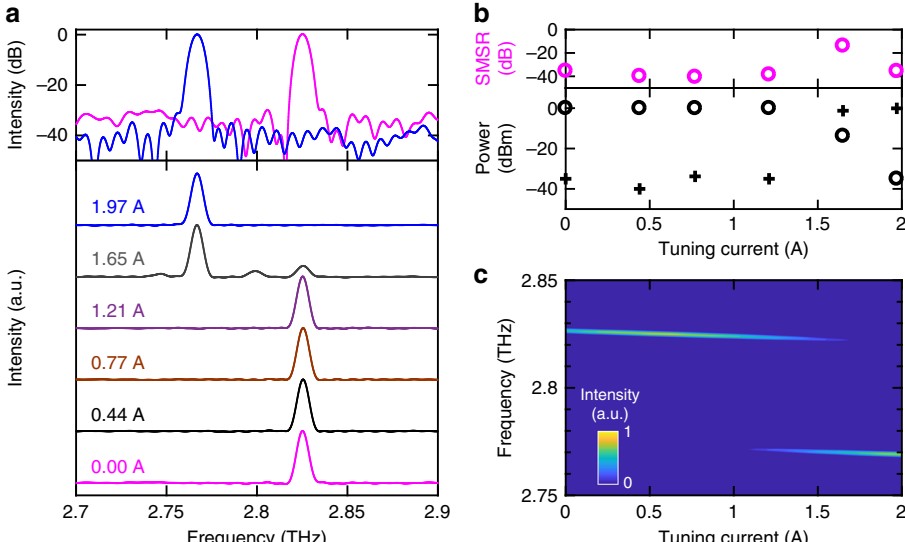

**Fig. 1** Intensity measurements from the coupled-cavity laser. **a** Steady-state emission spectra obtained experimentally using a Fourier transform infrared spectrometer with bolometric detection, at a heat sink temperature of 5 K. The laser cavity was driven at peak output power with quasi-direct current pulses with amplitude 0.75 A. The current supplied to the tuning cavity was varied, but kept below threshold, and acts as a localised heating element. Inset: Single mode emission with a side mode suppression ratio (SMSR) of ~35 dB is obtained at two discrete emission frequencies for tuning currents 0 A (magenta) and 1.97 A (blue). **b** Top: Experimental variation of the SMSR as a function of tuning current. Bottom: The corresponding output power of the two modes at 2.825 THz (circle) and 2.765 THz (cross). **c** Simulation of the steady-state emission spectrum of the coupled-cavity laser for different currents supplied to the tuning section, calculated using a transfer matrix model

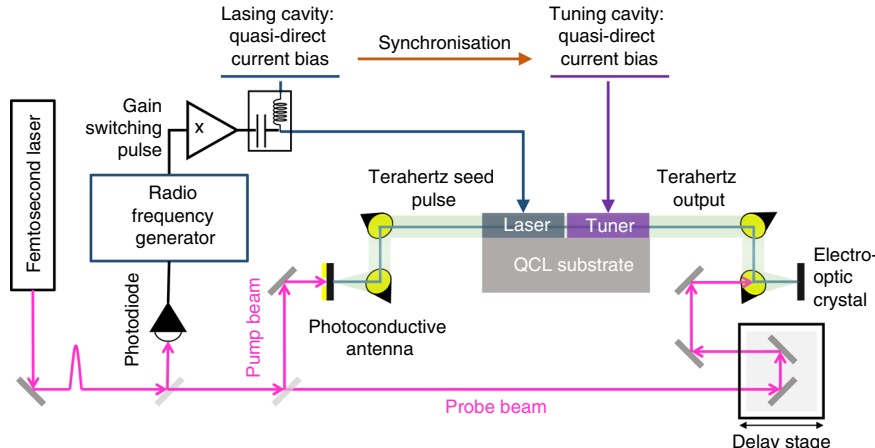

**Fig. 2** Illustration of the experimental arrangement used for injection seeding of the coupled-cavity laser. A Ti:Sapphire laser was used to generate femtosecond pulses, a fraction of the power was used to trigger a radio frequency source to generate gain switching pulses. The remaining power was split to form a pump beam and a probe beam. The pump beam was focused onto a photoconductive antenna to generate broadband terahertz pulses. These terahertz pulses were used to injection seed the lasing section of the coupled-cavity laser. The terahertz field emitted from the facet of the tuning section was sampled using an electro-optic crystal with the time-delayed probe beam. The gain switching pulses were combined with quasi-direct current pulses using a bias tee to drive the lasing cavity above threshold. A second quasi-direct current pulse generator was synchronised with the first pulse generator and was used to drive the tuning cavity

emission is seeded, fast gain-switching pulses were synchronised with the seed pulses such that the arrival of each seed pulse coincided with the laser being driven above lasing threshold. As such, the short seed pulse acts to synchronise the emission of the terahertz laser with the repetition frequency of the femtosecond laser used for coherent sampling. The emission from such seeded lasers has been simulated in refs. [21,22], both immediately following pulse injection and in the steady state. These studies have found the laser emission to be largely insensitive to changes in the pulse spectrum and pulse amplitude[21]. This injection seeding arrangement allows amplification of the seed pulse by the QCL gain medium, thereby seeding the laser so that the emission from

the opposite facet of the device can be sampled coherently, with ~70 fs resolution, using electro-optic sampling (Fig. 3a). The seed pulse is amplified by the gain medium as it propagates inside the coupled-cavity, with both partial reflection and transmission of the pulse occurring at each facet of the device as well as at the interfaces of the air gap dividing the cavity sections. A steady-state standing wave is thus established in the cavity after gain clamping, after a number of round trips (RT) of the cavity.

In our device, the RT time is calculated to be ~115 ps and the rise time of the gain-switching pulse is ~0.5 ns (i.e. ~4 RT). The variation in refractive index, reflectivity and mirror losses during the pulse rise time results in a delay in the amplification of the

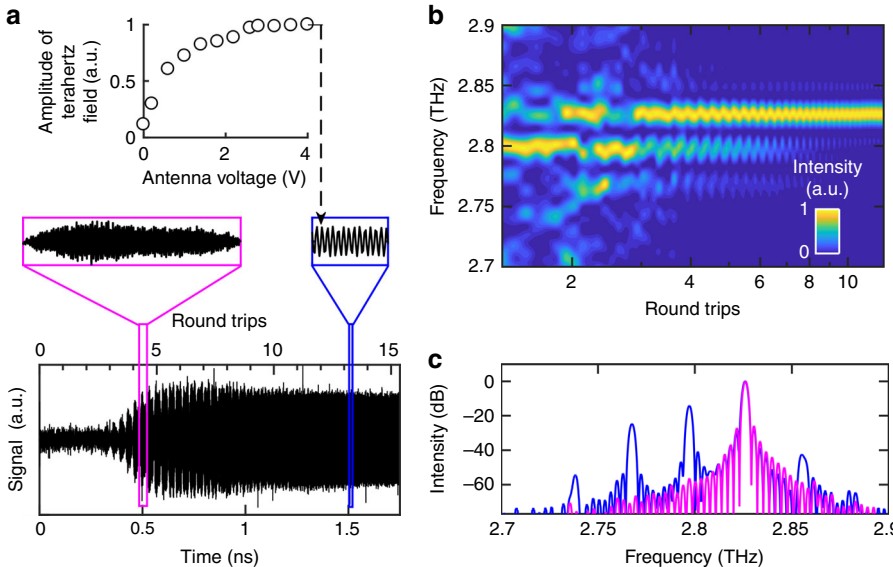

**Fig. 3** Coherent time-domain measurement of the electric field emitted from the coupled-cavity laser with the tuning cavity switched off. **a** The electric field emitted from the output facet of the tuning cavity, measured as a function of time. The seed pulse enters the laser cavity at $t = 0$. The lasing cavity is driven using quasi-direct current pulses of amplitude 0.65 A and the tuning cavity is switched off. Inset: Close-up of the measured electric field and the variation of the amplitude of the terahertz field at $t = 1500$ ps as a function of voltage applied across the broadband photoconductive antenna. An increase in the terahertz field was measured with respect to antenna bias, before saturating at an antenna bias >2.5 V implying that the emission is fully injection seeded by the seed pulse. **b** Plot showing the dynamic variation of the emission spectra, obtained from the fast Fourier transform of the time-domain electric field using a moving time window of width 150 ps. The x-axis plots the beginning of this time window on a logarithmic scale to emphasise the presence of mode hopping during the initial few round-trips in the cavity. **c** Emission spectra obtained from the fast Fourier transform of the time-domain electric field using a ~460-ps-wide time window beginning at $t = 400$ ps (blue) and at 1700 ps (magenta)

seed pulse. Indeed, the electric field experiences amplification only after ~340 ps (~3 RT), as can be seen from Fig. 3a. Additionally, due to the propagation time of light in the coupled cavity, the effective mirror losses can be treated as a time-dependent parameter. This temporal variation was calculated from the frequency dependence of the mirror losses arising due to the coupled-cavity geometry (see Methods), and was predicted to reach a steady-state within ~1.5–2 RT after the stabilisation of the gain-switching pulses. As a result, gain clamping and a steady-state electric field were observed after both the gain-switching pulses and the mirror losses reached a steady-state after ~800 ps (~7 RT). Interestingly, a periodic modulation of the field amplitude, suggesting multi-mode emission[17], is observed from ~340 to 800 ps (3−7 RT) before a stable single-mode oscillation developed, indicating a temporal evolution from multi-mode to single-mode emission (Fig. 3a, inset). The gradual decrease in the amplitude of the field after gain clamping could be caused by the shape of the gain-switching pulse, a loss of coherence or the effect of slower intra-pulse thermal stabilisation[23]. The amplitude of the emitted terahertz field from the laser was also measured at steady-state emission, i.e. $t > 1.5$ ns by varying the bias applied across the broadband antenna (Fig. 3a, inset). The terahertz field from the QCL was observed to saturate at antenna bias >2.5 V, indicating that the emission is fully injection seeded by the incident broadband terahertz pulses.

To verify this temporal evolution of the emission spectrum after switch-on, fast Fourier transforms were performed on a moving window of width 150 ps (~1.3 RT) to form a spectrogram. This time window was chosen such that the Fabry–Pérot modes in the laser cavity could be resolved spectrally, while still allowing their temporal evolution to be examined with sufficient time resolution. Fig. 3b shows the results of this analysis, which reveals hopping between modes at ~2.765, 2.795 and 2.825 THz within

the first 5 RT, before single-mode emission at the steady-state frequency of ~2.825 THz is established after ~10 RT. Additionally, although the power decreases after 3 RT for all modes except 2.825 THz, emission is still multi-mode within the first ~7–8 RT and corresponds to the periodic modulation of the electric field observed over similar time scales in Fig. 3a.

In order to improve the spectral resolution, a wide fast Fourier transform window of width 460 ps (~4 RT) was also applied to the time-domain data, averaging the emission spectra over this time window. Multiple emission modes are observed when the sampling window begins at $t = 400$ ps, which evolve into single mode emission at ~2.825 THz when the sampling window begins at $t = 1700$ ps (Fig. 3c). Furthermore, the frequency spacing between the modes is consistent with the longitudinal mode spacing expected in the lasing cavity (~29 GHz). This further substantiates a temporal evolution of the emission spectrum after switch-on of the laser.

The dynamic hopping between cavity modes after switch-on was also analysed by integrating the spectral power contained in each of the Fabry–Pérot modes, normalised to the total power. The time-dependence of the power of the modes at 2.795, 2.825 and 2.855 THz are shown in Fig. 4a. The power contained within the 2.825 THz mode was found to increase over times spanning from $t$~3−9 RT, corresponding to the amplification of the electric field over similar time scales (Fig. 3a). This coincides with the monotonic decrease in power of all other modes, although the mode at 2.795 THz is dominant for $t < 3$ RT. We believe that the mode hopping observed between $t$~1 and 4 RT arises from the increase in voltage supplied to the lasing cavity due to the rise time of the gain-switching pulse, and the resulting Stark shift of the gain. The monotonic decrease of power of all modes except 2.825 THz, over the next 2–3 RT, can be explained by the temporal variation of the effective mirror losses before attaining a steady state.

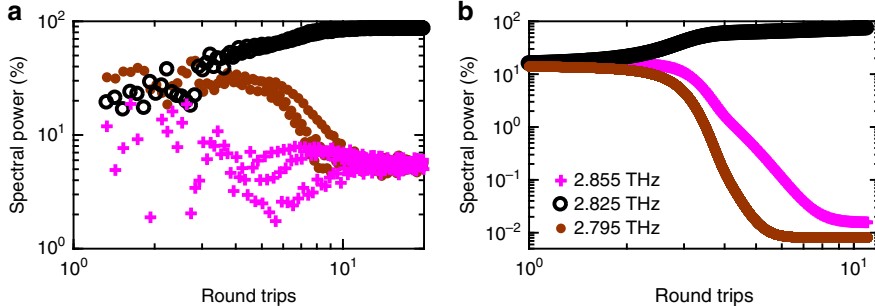

**Fig. 4** Distribution of power among lasing modes as a function of cavity round-trips. The power of three modes (2.795, 2.825 and 2.855 THz) normalised to represent percentage of total emitted power: **a** calculated from experimental data and **b** simulated using the reduced rate equation model. In each case, only the three dominant modes are shown for clarity

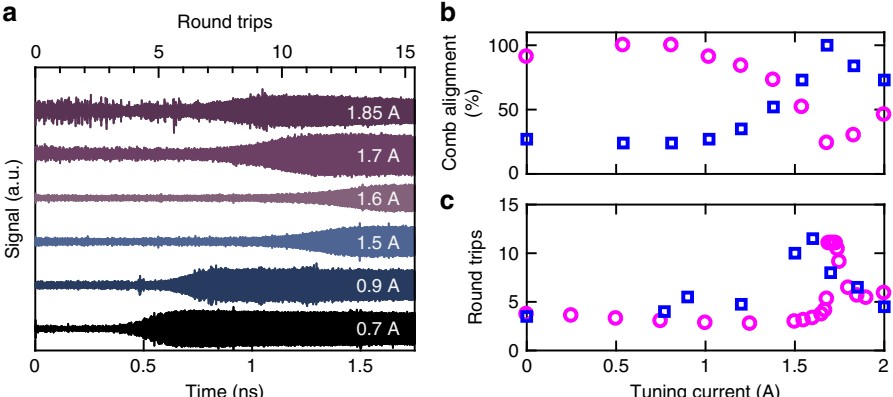

**Fig. 5** Coherent measurement of the electric field emitted from the coupled-cavity laser for different currents supplied to the tuning cavity. **a** Electric field emitted from the output facet of the tuning cavity, measured as a function of time for different currents supplied to the tuning section as indicated (offset for clarity). The lasing cavity is driven using quasi-direct current pulses of amplitude 0.65 A (the noise apparent in the 1.85 A trace arose from external vibrations). **b** Comb alignment parameter calculated using the transfer matrix model at frequencies 2.765 THz (blue) and 2.825 THz (magenta).
**c** Experimentally measured switch-on delay (blue) and laser stabilisation time (magenta) simulated using the reduced rate equation model (expressed as number of round trips), as a function of tuning current

To understand this dynamic selection from multi-mode to single-mode operation, the temporal redistribution of power between the Fabry–Pérot modes in a free-running terahertz coupled-cavity QCL was modelled using the reduced rate equation model (see Methods). A multimode switch-on delay of ~1.5–2 RT was simulated, followed by a mode competition in the next ~3–7 RT and eventually a steady-state single mode emission. The simulations predict that the power of all Fabry–Pérot modes except the mode at 2.825 THz decrease monotonically for $t > 4$ RT (Fig. 4b). Despite the model assuming a free-running laser, rather than an injection-seeded laser, the simulation results agree well with the overall behaviour observed in the experimental data, as well as those reported in refs. [14,20]. The discrepancies observed between the experimental and simulated temporal evolution could be explained by an underestimation of the gain bandwidth in the model arising due to fluctuations during the growth (see Methods).

Although the switch-on dynamics were measured at a constant 10 K, previous simulations indicate that the thermal-dependence of switch-on delay in terahertz QCLs is relatively weak at low temperatures[24], and that this remains the case until the power output begins to decay due to thermal effects (~50 K in the present case). This implies that the results presented here will remain valid over a comparable temperature range. Beyond this limit, Joule heating in the laser may also affect the switch-on delay.

Additionally, we note that the observed evolution from multi-mode to single mode emission is qualitatively similar to that simulated for a coupled-cavity diode laser[14], albeit occurring over timescales dependent on the carrier lifetimes of the laser gain material. As such, whereas such switch-on dynamics occur at nanosecond-timescales in diode lasers, here such processes occur at picosecond-timescales due to the shorter carrier lifetimes in terahertz QCLs. These results reveal how the transient multi-mode emission evolves into a steady-state single mode emission in a monolithic frequency-selective semiconductor laser.

An important feature of multi-section monolithic lasers is the ability to tune the emission frequency, which is controlled by changing the frequency at which modes of the respective frequency combs align. From this alignment condition, a further change in the refractive index of one or more cavity sections will cause a misalignment of the combs, resulting in a degradation of the side mode suppression ratio, before a mode hop occurs when the next alignment is reached, as shown in Fig. 1. As such, a study of the switch-on dynamics under these different conditions of comb alignment is important to understand the fundamental properties of such lasers, such as laser stabilisation, jitter and frequency cross-talk.

**Variation of laser switch-on dynamics with tuning.** In order to evaluate the effect of comb alignment on the switch-on dynamics of the coupled-cavity laser, the emitted electric field was also

recorded when a quasi-direct current pulse was applied to the tuning cavity. As shown in Fig. 5a the switch-on delay[7] was found to increase as the current was increased from 0 to 1.6 A, corresponding to a progressive misalignment of the frequency combs and an increase in the effective mirror losses at 2.825 THz. In addition, the switch-on delay was found to decrease after the mode switched to 2.765 THz, at tuning current amplitudes >1.6 A. Since the laser threshold and the gain-switching pulse rise time are the same as for the measurements shown in Fig. 3a, this change in the switch-on delay is attributed to a thermally induced change of the refractive index in the tuning cavity, which causes an increasing alignment of comb modes at 2.765 THz and a corresponding misalignment at 2.825 THz. Although the switch-on delay in terahertz QCLs has been predicted to increase with an increase in heat sink temperature[24], a decrease in switch-on delay is observed here at higher lattice temperatures, i.e. at tuning currents >1.6 A. This indicates a clear influence of the frequency comb alignment on the switch-on delay in such lasers.

The relationship between the comb alignment and the switch-on delay was further verified from the laser stabilisation time, i.e. the time required to attain a steady state, simulated using the transfer matrix model and the reduced rate equation model. For each tuning current, the comb alignment was calculated from overlap of the combs of the lasing and tuning cavities at 2.825 and 2.765 THz, normalised with respect to that obtained from a full comb alignment (Fig. 5b). The laser stabilisation time is plotted, in units of number of round trips, as a function of the tuning current in Fig. 5c. In agreement with the experimental observations, the longest stabilisation time is predicted at tuning currents close to the current at which the mode hops from 2.825 to 2.765 THz (~1.6 A), which corresponds to a switch in the alignment of frequency combs (Fig. 5b), and also the worst side mode suppression observed in Fig. 1c. Furthermore, while the side mode suppression ratio obtained experimentally are similar (~35–40 dB) at tuning currents other than 1.6 A, the laser stabilisation time is found to be more sensitive to the comb alignment and varies between 7 and 13 RT through the tuning range (Fig. 5a). These observations suggest that although both the steady-state spectral behaviour and the temporal switch-on dynamics of mode selection depend on the alignment of the frequency combs, the temporal switch-on time is more sensitive to the comb alignment.

The increase in the switch-on delay of ~8 RT (~920 ps) due to change in the alignment of comb modes is almost an order of magnitude larger than that simulated due to an increase in the temperature below the power roll-off[24]. Moreover, these changes in switch-on delay occur at larger time scales than the carrier lifetime in terahertz QCLs. As such, the variation in the switch-on dynamics due to changes in the alignment of comb modes are expected at higher temperatures since these effects arise due to the coupled-cavity geometry of the devices.

## Discussion

We have measured the ultrafast switch-on dynamics and dynamic frequency selection from transient multi-modes into a steady-state single mode emission in a monolithic frequency-tuneable semiconductor laser using coherent time-domain sampling of the laser emission. While a terahertz QCL based on intersubband transition of electrons, with picosecond carrier lifetimes, was used for these experiments, our observations are qualitatively similar to the transient behaviour simulated in coupled-cavity diode lasers over nanosecond-timescales in ref.[14]. We note that compared to a single cavity laser, the transient dynamics observed here are due to the frequency-dependent variation in mirror losses, which

originate due to the coupled-cavity design. As such, these observations are qualitatively applicable to any multi-cavity frequency-tuneable laser, such as lasers with sampled grating distributed Bragg reflectors, where frequency selection is determined by comb alignment. However, the timescales of such transient dynamics would depend on the carrier lifetimes of the laser gain material. Similarly, the timescales for the switch-on dynamics within a terahertz QCL are likely to be sensitive to the materials and active region design, although these can be predicted through simulation of the carrier dynamics[24].

These results suggest that for any frequency-tuneable laser based on Vernier selection, where switching speeds are important, the variation of switch-on time should be considered. For example, when fast switching between emission frequencies is required, care should be taken to design the laser cavity such that regimes of intermediate comb alignment are avoided.

In conclusion, we report dynamic hopping between lasing modes and the temporal evolution of a transient multi-mode emission into a steady-state single mode emission in a coupled-cavity QCL. We also found that, whereas hopping between single modes can be achieved through a small change in tuning current, the stabilisation time in such lasers is more sensitive to the alignment between the frequency combs supported by each cavity section.

## Methods

**Fabrication**. A terahertz frequency QCL based on a conventional bound-to-continuum active region design[25] was grown using molecular beam epitaxy. Growth started with a 250-nm-thick GaAs buffer layer grown on a semi-insulating GaAs substrate. A 300-nm-thick $Al_{0.5}Ga_{0.5}As$/GaAs etch stop layer was grown next, followed by a 700-nm-thick $n$-doped GaAs layer doped with Si at $2 \times 10^{18}$ cm$^{-3}$, forming a buried contact layer. Alternating layers of $Al_{0.15}Ga_{0.85}As$ forming the active region stack were then grown in a sequence starting from the injection barrier *3.8*/11.6/*3.5*/11.3/*2.7*/11.4/*2*/12/*2*/12.2/*1.8*/12.8/*1.5*/15.8/*0.6*/9/*0.6*/14 nm (barriers in italics). The 11.4 and 12-nm-wide wells (underlined) were $n$-doped with Si at $3.2 \times 10^{16}$ cm$^{-3}$. We note that both spatial and temporal variations in flux densities during the growth resulted in a slight variation from the heterostructure design in experimental devices The sequence was repeated 90 times and the growth concluded with an 80-nm-thick $n$-doped GaAs layer, with Si doping density $5 \times 10^{18}$ cm$^{-3}$.

Terahertz QCL devices were fabricated with semi-insulating surface plasmon ridge waveguides, with the coupled cavities formed by focused-ion-beam etching. Laser ridges with widths of 150 μm and thickness of 11.6 μm were defined using optical photolithography and wet chemical etching using an aqueous etchant solution of $H_2SO_4$, $H_2O_2$ and $H_2O$, premixed in the ratio 1:8:40. A eutectic alloy of Au/Ge/Ni was deposited using vacuum thermal evaporation to form Ohmic contacts to the laser. Similarly, cladding layers of Ti/Au were evaporated to form the waveguide. The substrate was thinned to a thickness of ~200 μm and a soldering layer of Ti/Au was evaporated on the thinned substrate. A 4.81-mm-long device was cleaved and soldered onto a Cu block using indium. The coupled-cavity was formed after packaging using a focused-ion-beam milling technique to etch a 12-μm-deep air gap, thereby splitting the cavity into a lasing and a tuning cavity. For this study, a QCL device was fabricated with a tuning cavity of length 3.43 mm and a lasing cavity of length 1.38 mm, which were separated by an air gap of 13 μm. Finally, the top of the lasing and the tuning cavities were wire-bonded to two separate ceramic pads allowing independent electrical connections to the cavities.

**Modelling**. In order to simulate the dynamic behaviour of the terahertz QCL a hybrid model was developed based on a transfer matrix model combined with a multi-mode reduced rate equation model. The air gap and the passive cavity were modelled as an effective mirror with frequency-dependent reflectivity. In this way the coupled-cavity laser could be described as a single cavity with an effective complex mirror, and could thus be modelled using the following transfer matrix based on scattering matrix theory:[14,16,20]

$$H(\nu) = \frac{t_a t_p S_{s21} \sqrt{(1 - r_4^2)(1 - r_1^2)}}{1 - (S_{s12}^2 - S_{s11}^2)r_1 r_4 t_a^2 t_p^2 - S_{s11} r_4 t_p^2 - S_{s22} r_1 t_a^2}, \quad (1)$$

where $t_a$ and $t_p$ are the single pass transmission coefficients in the lasing and the tuning cavities respectively, $S_{s11}$, $S_{s12}$, $S_{s21}$ and $S_{s22}$ are the scattering coefficients, and $r_1$ and $r_4$ are the reflection coefficients at the end facets of the laser. The eigenmodes of the coupled-cavity laser ($\nu_m$) were calculated by setting the denominator of the transfer function to be zero.

**Table 1 Quantum cascade laser parameters used in reduced rate equation based numerical simulations**

| Symbol | Description | Value/definition |
|--------|-------------|------------------|
| $\nu_m$ | Eigenmode frequency for mode $m$ | 2.705, 2.735, 2.765, 2.795, 2.825, 2.855, 2.885 THz |
| $\eta_3$ | Injection efficiency into upper laser level | 54.41% |
| $\eta_2$ | Injection efficiency into lower laser level | 1.65% |
| $G_p$ | Peak gain of the gain spectrum | $5.3 \times 10^4\,\text{s}^{-1}$ |
| $M$ | Number of repetitions of the active module | 90 |
| $\beta_{sp}$ | Spontaneous emission factor | $1.627 \times 10^{-4}\,\text{s}^{-1}$ |
| $\tau_3$ | Total carrier lifetime in upper laser lifetime | $5 \times 10^{-12}\,\text{s}$ |
| $\tau_2$ | Total carrier lifetime in lower laser lifetime | $2.1 \times 10^{-11}\,\text{s}$ |
| $\tau_{32}$ | Non-radiative relaxation time from upper laser level to lower laser level | $1.76 \times 10^{-10}\,\text{s}$ |
| $\tau_{sp}$ | Spontaneous emission lifetime | $5 \times 10^{-10}\,\text{s}$ |

The eigenmodes calculated from this single cavity effective mirror model were then used to establish a dynamic model for the terahertz QCL based on the multi-mode reduced rate equation approach. This complete dynamic model describes the interactions of carrier densities in the upper and lower laser levels, photon densities at each eigenmode frequencies, and the lattice temperature in the passive cavity. To this end, the terahertz QCL active region was modelled using a full Schrödinger–Poisson energy balance rate equation simulation. The effect of Joule heating on the QCL lattice temperature, and the resulting variation in carrier transport via changes of scattering rates, electron subband lifetimes and electronic temperatures, were included in this model[26,27]. This allowed the static electron transport parameters that vary with lattice temperature and driving current to be determined at the eigenmode frequencies calculated from the transfer matrix model. These results from the Schrödinger–Poisson model were then combined with the reduced rate equations to form a hybrid model that was used to simulate the time-dependent photon and electron densities of the laser levels at the coupled-cavity eigenmode frequencies[20,28]. The multi-mode reduced rate equations used here is described by the following coupled equations:

$$\frac{dN_3(t)}{dt} = \frac{\eta_3 I_a(t)}{q} - (N_3(t) - N_2(t))\sum_{m=1}^{N} G_m S_m(t) - \frac{N_3(t)}{\tau_3}, \quad (2)$$

$$\frac{dN_2(t)}{dt} = \frac{\eta_2 I_a(t)}{q} + (N_3(t) - N_2(t))\sum_{m=1}^{N} G_m S_m(t) + \frac{N_3(t)}{\tau_{32}} + \frac{N_3(t)}{\tau_{sp}} - \frac{N_2(t)}{\tau_2}, \quad (3)$$

$$\frac{dS_m(t)}{dt} = M(N_3(t) - N_2(t))G_m S_m(t) - \frac{S_m(t)}{\tau_{pm}} + \frac{M\beta_{sp}N_3(t)}{\tau_{sp}},$$
$$\text{where } m = 1, 2, \dots, N \quad (4)$$

$$\frac{dT_p(t)}{dt} = \frac{1}{m_p c_p}\left[I_p(t)V_p(t) - \frac{T_p(t) - T_0}{R_{th}}\right], \quad (5)$$

where $N_3(t)$ and $N_2(t)$ are the carrier population in the upper and lower laser levels in the QCL, $q$ is the electronic charge, $G_m$ and $S_m(t)$ are the gain factor and the photon population in eigenmode $m$ respectively, $I_a(t)$ and $I_p(t)$ are the current supplied to the lasing and the tuning cavity, $M$ is the number of active periods in the QCL, $\tau_{pm}$ is the photon lifetime for mode $m$, $T_p(t)$ is the lattice temperature in the tuning cavity, $T_0$ is the cold finger temperature, $m_p$ and $c_p$ are the mass and specific heat capacity of the chip, $V_p$ is the voltage at the tuning cavity and $R_{th}$ is the thermal resistance. The definition and values of all other parameters used in Eqs. (2)–(5) are provided in the Table 1. The variation of refractive index due to Joule heating in the tuning section, and the resulting mode tuning dynamics, was also included in this reduced rate equation model. The experimental rise time of the gain-switching pulses was also included in the model to assess the laser switch-on times accurately. The model was solved using a fifth-order Runge–Kutta method[20], and the optical power at each eigenmode is calculated using the following expression:

$$P_m(t) = \frac{\eta_m h\nu_m S_m(t)}{\tau_{pm}}, \quad (6)$$

where $\eta_m$ is the power out coupling coefficient for mode $m$.

The transfer matrix model was used to calculate the eigenmode frequencies of the coupled-cavity laser in the steady state and the frequency-dependent variation of the mirror losses. To analyse the transient behaviour a time-dependent mirror loss was calculated through fast Fourier transform of the frequency-dependent mirror loss. Here we have assumed an instantaneous rise time for simplicity. Our results indicate that the mirror losses are periodic with ~0.5 RT, and decay exponentially with a time constant of ~0.5 RT.

The reduced rate equation model was solved to simulate the switch-on dynamics in the coupled-cavity QCL. The temporal evolution of a multimode emission to a single mode emission was simulated at tuning cavity current in the range 0–2 A.

**Measurement.** The device was mounted in a continuous-flow helium-cooled cryostat and cooled to a temperature <10 K. The lasing section was driven by a train of 5-μs-long current pulses at a repetition rate of 10 kHz, with the tuning section being driven with wider 10 μs current pulses below the lasing threshold (2.1 A). These wider pulses served to introduce a controlled index perturbation in the tuning section through localised Joule heating, as described in ref. [16]. Thermal conduction between the lasing and the tuning cavities, through the monolithic substrate, was ignored due to the low duty cycle (10%) of the tuning pulses. The free-running steady-state emission spectra of the device were measured, with a resolution of 7.5 GHz, using a Brucker IFS66/V Fourier transform infrared spectrometer employing a cryogenically cooled Ge:Ga bolometric detector.

The temporal dynamics of the laser emission were characterised using an injection seeding technique, similar to that described in ref. [17]. A Ti:Sapphire laser, operating at ~800 nm, was used to generate 100 fs pulses at a repetition rate of 76.47 MHz, which were focused onto a biased GaAs photoconductive antenna. The resulting recombination of free carriers with holes generated broadband terahertz pulses centred at ~1.27 THz and with a bandwidth of ~3 THz, and were used to injection seed the coupled-cavity lasing cavity. The gain-switching pulses (with a rise time ~0.5 ns) were generated using a pulse generator triggered by pulses from the Ti: Sapphire laser. These pulses were offset with quasi-direct current (5-μs-long) pulses using a bias tee, which together drive the laser above threshold. This arrangement allowed synchronisation between the pulses driving the lasing cavity above threshold with the arrival of the injection seeding pulses from the broadband antenna. Unlike the shorter lasing cavity, the longer tuning cavity was driven with only quasi-direct current (10-μs-long) current pulses below lasing threshold. The terahertz field emitted from the facet of the tuning section was sampled using an electro-optic sampling arrangement with a time-delayed probe beam from the Ti:Sapphire laser.

**Code availability.** The computer codes for the Schrödinger–Poisson scattering transport simulations are openly available at https://code.launchpad.net/qwwad. The computer codes for the reduced rate equation model are available from the corresponding author upon request.

**Data availability.** Research data associated with this paper are openly available from the University of Leeds Research Data Repository (https://doi.org/10.5518/163)

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

## Acknowledgements
We acknowledge the support from: the Engineering and Physical Sciences Research Council (EPSRC), UK ('COTS' and 'HyperTerahertz' programmes, EP/J017671/1 and EP/P021859/1); European Union FET-Open grant ULTRAQCL 665158; the European Cooperation in Science and Technology (COST) Action BM1205; Centre National de la Recherche Scientifique (CNRS), France; and a Royal Society International Exchange grant (IE120898). E.H.L. and A.G.D. are grateful for support from the Royal Society and Wolfson Foundation. We also acknowledge support from the Australian Research Council's Discovery Projects Funding Scheme (Grant DP 160 103910).

## Author contributions
I.K. and S.S.D. conceived and designed the experiments. E.H.L. and L.H.L. grew the QCL using molecular beam epitaxy. I.K. fabricated the device. I.K., F.W. and H.N. conducted the experiments. I.K., X.Q., A.V., G.A. and A.T.G. designed the simulations. I.K., S.S.D., P.D., J.R.F., J.M., T.T. and A.D.R. interpreted the results. S.S.D., E.H.L., A.G.D., J.E.C., D.I., A.D.R. and J.T. supervised the project. The manuscript was written by I.K., J.R.F., P.D., S.S.D. and A.G.D., and edited by all authors.

## Additional information

**Competing interests:** The authors declare no competing interests.

