## [Peer Review File · Nature Communications]

Reviewers' comments:

Reviewer #1 (Remarks to the Author):

Authors present the first report of coherence measurements of dynamic mode selection and switch-on processes in a monolithic single-mode frequency tunable semiconductor laser. The quantum cascade laser (QCL) structure operating at 2.8 THz is very similar to a cleaved-coupled-cavity (C3) used long time ago with interband diode lasers. The device under study is made with one section electrically driven above threshold (lasing cavity) whereas the other is electrically driven below to form a passive tuning cavity. The air gap between the two sections is fixed to 13 microns. The latter with the passive tuning cavity is equivalent to an effective mirror with frequency-dependent reflectivity and loss parameters. Lasing action takes place at discrete frequencies both supported by the lasing cavity and for which the effective mirror losses are a minimum. To tune the emission frequency, the refractive index of the tuning cavity is adjusted through Joule heating hence changing the frequency-dependence of the effective mirror losses, or the free spectral range of the longitudinal modes.

The main finding of the paper is the dynamic hopping between lasing modes and the temporal dynamics, which evolves from a transient multi-mode emission into a steady-state single mode emission. Such observations are confirmed using a coherent time-domain sampling of the laser emission based on an injection seeding technique. Authors claim that the observed switch-on dynamics is applicable to any type of frequency tunable lasers with or without complex cavities, such as sampled grating DBR lasers or even DFBs in which similar mode competition is expected. Experimental results are also fully confirmed by numerical computations using both transfer matrix method and rate equations.

Although I am quite skeptical with the main conclusions, I think the paper meets the standards of Nature Communications. To this end, I can support the publication after minor revisions.

1. Authors should give more details about the injection-seeding technique that was used for the phase-resolved measurements of the THz field. As any master-slave configuration, the QCL might become sensitive to the injection parameters hence leading to possible multiple types of nonlinear dynamics. In this context, authors may want to argue by using the following reference Optics Communications, Vol. 392, pp. 196-201, 2017.

2. Measurements were performed with Liquid-He at 4K so very far from room temperature. How sensitive is the injection seeding technique with temperature? Increasing the temperature will also result in speeding-up the carriers. Can we anticipate any modifications on the switch-on dynamics over temperature?

3. I am bit skeptical about the conclusions. Between a QCL and a diode laser, the carrier dynamics does not take place on the same timescale (sub-picosecond vs. sub-nanosecond). To me, this is the weakness of the paper. Authors claim that the switch-on dynamics remains valid whatever the laser structures but they did the measurements only with a THz QCL at 4K. They could have checked their conclusions by using another laser such as a simple telecom laser (C3, DFB or DBR whatever) operating at room temperature.

4. Can the switch-on dynamics be influenced by material parameters like the linewidth enhancement factor of the THz QCL? If so, please comment.

5. Authors mentioned "The switching dynamics and mode selection processes may play a limiting role in monolithic frequency tunable lasers used in telecommunications and for non-linear four wave mixing, and may limit the switching time or result in mixing of multiple modes on ultrafast timescales". For a publication in Nature Communications, authors should give broader insights and perspectives on what to be done to improve frequency tunable lasers.

Reviewer #2 (Remarks to the Author):

This manuscript reports measurements on the switch-on dynamics of a coupled-cavity THz QCL, using an established technique (used before by some of the authors) of injection seeding a QCL followed by subsequent coherent measurement of the laser electric field using THz time-domain spectroscopy techniques. The novelty of this paper then is the coupled-cavity THz QCL that the technique is applied to. Such lasers are of interest because of their use as tunable lasers using Vernier effect of the Fabry-Perot modes of the two cavities as a bias current is applied to the "tuning" cavity segment.

The paper is well written and the experimental data quality is excellent. The measured dynamics seem to roughly match the simulation data. A particularly interesting phenomenon was observed – the switch on stabilization time is much longer in the region when the CC switches from one preferred mode to the other.

I recommend acceptance with minor revisions.

Two complaints it would be appreciated if the authors could address:

- In an effort to broaden the appeal of the paper, the authors have claimed that these results are directly applicable to any coupled/cavity semiconductor laser. I am not immediately convinced – QCLs have much different dynamics than interband lasers, with picosecond lifetimes rather than nanosecond. More must be clarified on this point – or else the authors should remove these claims.
- In general the paper could use more physical and intuitive explanation of the models and the results. While there is convergence between experimental and modelling results, almost all details of the modeling have been offloaded to another paper. Similarly, discussion of the lasers themselves and their behavior have been previously reported. This makes it difficult to discuss the results in much depth – particularly when discussing the switch on time dynamics.

Reviewer #1 (Remarks to the Author):

- 1. Authors should give more details about the injection-seeding technique that was used for the phase-resolved measurements of the THz field. As any master-slave configuration, the QCL might become sensitive to the injection parameters hence leading to possible multiple types of nonlinear dynamics. In this context, authors may want to argue by using the following reference *Optics Communications*, Vol. 392, pp. 196-201, 2017.**

We can clarify for the reviewer that our measurement methods are same as those used in Ref. [17] of the manuscript, which gives a full account of the injection seeding technique. We have also verified that the QCL emission is locked fully to the seed pulse by measuring the emission of the QCL at steady state (i.e. at $t > 1.5$ ns) as a function of the injected THz pulse amplitude, similar to the techniques described in <https://doi.org/10.1364/OL.37.000731>.

Accordingly, we have added a note in the main text (section 2, page 6).

“As such, the short seed pulse acts to synchronise the emission of the QCL with the repetition frequency of the femtosecond laser used for coherent sampling.”

We have also expanded the methods section of the manuscript to describe the experimental technique in more detail, and have also included seed pulse saturation data as Supplementary Information. We have added the following text to the Methods (Measurement) section (pages 17–18):

“The amplitude of the emitted THz field from the CC THz QCL was measured at steady state emission, i.e. $t > 1.5$ ns by varying the bias applied across the broadband antenna (see Supplementary Information). The THz field from the QCL was observed to saturate at antenna bias > 2.5 V, indicating that the emission is fully injection seeded by the incident broadband THz pulses.”

The reviewer raises an interesting point in his/her query concerning how emission from the QCL depends on the injection parameters. In fact, this subject has already been investigated in Freeman *et al. Phys. Rev. A*, 87, 063817 (2013) as well as the *Optics Communications* article suggested by the reviewer. Figure 5 from Freeman *et al.* simulates the emission from an injection-seeded QCL, both immediately following pulse injection and in the steady-state, for different injection parameters. This previous study concludes that the QCL emission is largely unchanged when either the spectrum or the amplitude of the seed pulse is changed. We have added a note in the main text (section 2, page 6):

“The emission from such seeded QCLs has been simulated in Refs. 21,22, both immediately following pulse injection and in the steady-state. These studies have found the QCL emission to be largely insensitive to changes in the pulse spectrum and pulse amplitude.²¹”

- 2. Measurements were performed with Liquid-He at 4K so very far from room temperature. How sensitive is the injection seeding technique with temperature?**

In the injection seeding technique used here, the broadband THz seed pulses were generated at room temperature. As such, the generation of seed pulses is invariant in all measurements. The refractive index (and hence reflectivity) of the QCL is affected only weakly by temperature, and as such the coupling of the pulse into the cavity will not vary significantly. Previous simulations (Ref. [24]) also show that the carrier dynamics of a QCL exhibit weak thermal dependence until a threshold temperature is reached (~ 50 K in the present work), above which the laser output power is observed to decay rapidly. As such, we are confident that the injection seeding will be temperature invariant within the normal operating range of the QCL.

Increasing the temperature will also result in speeding-up the carriers. Can we anticipate any modifications on the switch-on dynamics over temperature?

As noted above, simulations [24] indicate that the switch-on dynamics of THz QCLs show weak thermal dependence as long as the device does not exceed the threshold temperature at which its power is observed to degrade. Indeed, the ~ 920 -ps change in switch-on delay that we report in this work [see Fig. 5(a)] is much larger than the simulated thermal variation (~ 150 ps) in [24]. As such, we expect that the results presented in our manuscript would remain valid within the normal operating temperature range of the QCL (~ 50 K in the present case).

We agree with the reviewer that a large increase in operating temperature, i.e. beyond the power roll-off, would result in significant changes in switch-on delay due to a decrease in the carrier-injection efficiency. However, such high operating temperatures not only degrade the device performance, but would also change the tuning characteristics arising from a thermal perturbation of the refractive index and may even favour multi-mode emission.

To clarify this we have added the following text in section 2 (page 9):

“Although, the switch-on dynamics were measured at a constant 10 K, previous simulations indicate that the thermal-dependence of switch-on delay in THz QCLs is relatively weak at low temperatures,²⁴ and that this remains the case until the QCL power begins to decay due to thermal effects (~50 K in the present case). This implies that the results presented here will remain valid over a comparable temperature range. Beyond this limit, Joule heating in the QCL may also affect the switch-on delay.”

Section 3 (page 11):

“The increase in the switch-on delay of ~8 RTs (~920 ps) due to change in the alignment of comb modes is almost an order of magnitude larger than that simulated due to an increase in the temperature below the power roll-off.²⁴ Moreover, these changes in switch-on delay occur at larger time scales than the carrier lifetime in THz QCLs. As such, the variation in the switch-on dynamics due to changes in the alignment of comb modes are expected at higher temperatures since these effects arise due to the coupled-cavity geometry of the devices.”

- 3. I am bit skeptical about the conclusions. Between a QCL and a diode laser, the carrier dynamics does not take place on the same timescale (sub-picosecond vs. sub-nanosecond). To me, this is the weakness of the paper. Authors claim that the switch-on dynamics remains valid whatever the laser structures but they did the measurements only with a THz QCL at 4K.**

We agree that the timescales over which such selection occur are dependent on the carrier dynamics. Whereas the switch-on delay is experimentally observed here to be ~340 ps in the THz QCL due to intersubband transitions, the switch-on delay in a coupled cavity diode laser is simulated to be typically > 1 ns, as shown in Ref. [14] in the manuscript. However, the selection of steady-state single mode emission measured here is qualitatively similar to the effect simulated in a diode laser, albeit over different time scale due to intersubband carrier dynamics.

Furthermore, we have observed the switch-on delay in a THz QCL to vary as a function of comb alignment due to Vernier selection. This phenomenon is expected in any Vernier-effect based laser, although the timescales of such dependency would again be governed by the carrier lifetimes.

To clarify this issue we have added the following text in section 2 (page 9):

“Additionally, we note that the observed evolution from multi-mode to single mode emission is qualitatively similar to that simulated for a CC diode laser,¹⁴ albeit occurring over timescales dependent on the carrier lifetimes of the laser gain material. As such, whereas such switch-on dynamics occur at ns-timescales in diode lasers, here such processes occur at ps-timescales due to the shorter carrier lifetimes in THz QCLs.”

We have modified the conclusions (page 11) and have added the following text:

“Whilst a THz CC QCL based on intersubband transition of electrons, with ~ps carrier lifetimes, was used for these experiments, our observations are qualitatively similar to the transient behaviour simulated in CC diode lasers over ns-timescales in Ref. 14. We note that compared to a single cavity laser, the transient dynamics observed here are due to the frequency dependent variation in mirror losses, which originate due to the CC design. As such, these observations are qualitatively applicable to any multi-cavity frequency tuneable laser, such as sampled grating DBR lasers, where frequency selection is determined by comb alignment. However, the timescales of such transient dynamics would depend on the carrier lifetimes of the laser gain material.”

They could have checked their conclusions by using another laser such as a simple telecom laser (C3, DFB or DBR whatever) operating at room temperature.

We agree that this would be an interesting investigation. However, injection seeding of a telecom laser is technologically challenging and is beyond the scope of the current work. Since we are unable to experimentally measure a DFB laser exhibiting spatial hole burning and verify the effect of degraded SMSRs on temporal dynamics in such lasers, we have now removed this claim throughout the manuscript.

- 4. Can the switch-on dynamics be influenced by material parameters like the linewidth enhancement factor of the THz QCL? If so, please comment.**

The reviewer is correct in noting that material parameters may influence the switch-on dynamics. Unlike diode lasers though, QCLs are intersubband devices, with gain resulting from transitions between two subbands with parallel dispersion curves. This results in a spectrally symmetric gain profile, and a negligible linewidth-enhancement factor. As such, this particular parameter is not significant for a study of THz QCLs. However, the design of the QCL active region heterostructure would certainly affect the material gain, carrier lifetimes, and injection efficiency into the upper laser level. As discussed above though, the effect would remain qualitatively identical to the measurements presented here, with an adjustment in timescales.

We have added a note on this in the conclusions section (page 12):

“Similarly, the timescales for the switch-on dynamics within a THz QCL are likely to be sensitive to the materials and active-region design, although these can be predicted through simulation of the carrier dynamics.²⁴”

5. **Authors mentioned “The switching dynamics and mode selection processes may play a limiting role in monolithic frequency tunable lasers used in telecommunications and for non - linear four wave mixing, and may limit the switching time or result in mixing of multiple modes on ultrafast timescales”. For a publication in Nature Communications, authors should give broader insights and perspectives on what to be done to improve frequency tunable lasers.**

We thank the reviewer for this suggestion, and have elaborated further in the conclusion section (page 12) on the insights resulting from our works:

“These results suggest that for any frequency tuneable laser based on Vernier selection, where switching speeds are important, the variation of switch-on time with frequency tuning should be considered. For example, when fast switching between emission frequencies is required, care should be taken to design the laser cavity such that regimes of intermediate comb alignment are avoided.”

Reviewer #2 (Remarks to the Author):

1. **In an effort to broaden the appeal of the paper, the authors have claimed that these results are directly applicable to any coupled/cavity semiconductor laser. I am not immediately convinced – QCLs have much different dynamics than interband lasers, with picosecond lifetimes rather than nanosecond. More must be clarified on this point – or else the authors should remove these claims.**

We agree that the variation in timescales between different lasers was not given full consideration in the initial manuscript, and we have addressed this concern in the response to reviewer 1 (Question 3 and 4) above.

2. **In general the paper could use more physical and intuitive explanation of the models and the results. While there is convergence between experimental and modelling results, almost all details of the modeling have been offloaded to another paper. Similarly, discussion of the lasers themselves and their behavior have been previously reported. This makes it difficult to discuss the results in much depth – particularly when discussing the switch on time dynamics.**

We thank the reviewer for this useful suggestion. In order to provide more detailed explanation of the modelling approach and experimental details of the laser devices, we have made the following changes to the manuscript:

We have rearranged the following text in section 1 (page 5) and have added the highlighted text:

“The CC device was modelled, in the first instance, using a transfer matrix model based on scattering matrices to simulate the eigenfrequencies in the CC THz QCL. A dynamic reduced rate equation (RRE) model²⁰ was used to simulate the spectral power distribution amongst the different eigenfrequencies, with the carrier lifetimes having been obtained from an energy-balanced Schrödinger–Poisson scattering transport calculation. The RRE model includes the interaction between photon density and electron population for different subbands in the QCL and for multiple FP modes in the CC THz QCL, and also includes thermal effects. As such, it can simulate not only the steady-state tuning characteristics, but also temporal dynamics of mode selection. The numerical models are described in the Methods section.”

We have also added the following text in section 2 (pages 8–9):

“A multimode switch-on delay of ~1.5–2 RTs was simulated in the CC THz QCL, followed by a mode competition in the next ~3–7 RTs and eventually a steady-state single mode emission”

We have expanded the Methods (modelling) section (page 14) extensively, in order to provide a complete description of the numerical model, as noted below:

“In order to simulate the dynamic behaviour of the CC THz QCL a hybrid model was developed based on a transfer matrix model combined with a multi-mode RRE model. The air gap and the passive cavity were modelled as an effective mirror with frequency-dependent reflectivity. In this way the CC THz QCL could be described as a single cavity with an effective complex mirror, and could thus be modelled using the following transfer matrices based on scattering matrix theory:^{14,16,20}

$$H(\nu) = \frac{t_a t_p S_{s21} \sqrt{(1-r_4^2)(1-r_1^2)}}{1 - (S_{s12}^2 - S_{s11}^2) r_1 r_4 t_a^2 t_p^2 - S_{s11} r_4 t_p^2 - S_{s22} r_1 t_a^2} \quad (1)$$

where t_a and t_p are the single pass transmission coefficients in the lasing and the tuning cavities, respectively, S_{s11} , S_{s12} , S_{s21} and S_{s22} are the scattering coefficients, and r_1 and r_4 are the reflection coefficients at the end facets of the CC THz QCL. The eigenmodes of the CC THz QCL (ν_m) were calculated by setting the denominator of the transfer function to be zero.

The eigenmodes calculated from this single cavity effective mirror model were then used to establish a dynamic model for the CC THz QCL based on a multi-mode RRE approach. This complete dynamic model describes the interactions of carrier densities in the upper and lower laser levels, photon densities at each eigenmode frequencies, and the lattice temperature in the passive cavity.”

Pages 15–16:

“The multi-mode RRE used here is described by the following coupled equations:

$$\frac{dN_3(t)}{dt} = \frac{\eta_3 I_a(t)}{q} - (N_3(t) - N_2(t)) \sum_{m=1}^N G_m S_m(t) - \frac{N_3(t)}{\tau_3} \quad (2)$$

$$\frac{dN_2(t)}{dt} = \frac{\eta_2 I_a(t)}{q} + (N_3(t) - N_2(t)) \sum_{m=1}^N G_m S_m(t) + \frac{N_3(t)}{\tau_{32}} + \frac{N_3(t)}{\tau_{sp}} - \frac{N_2(t)}{\tau_2} \quad (3)$$

$$\frac{dS_m(t)}{dt} = M (N_3(t) - N_2(t)) G_m S_m(t) - \frac{S_m(t)}{\tau_{pm}} + \frac{M \beta_{sp} N_3(t)}{\tau_{sp}} \quad (4)$$

$m = 1, 2, \dots, N$

$$\frac{dT_p(t)}{dt} = \frac{1}{m_p c_p} \left[I_p(t) V_p(t) - \frac{T_p(t) - T_0}{R_{th}} \right] \quad (5)$$

where $N_3(t)$ and $N_2(t)$ are the carrier population in the upper and lower laser levels in the QCL, q is the electronic charge, G_m and $S_m(t)$ are the gain factor and the photon population in eigenmode m , $I_a(t)$ and $I_p(t)$ are the current supplied to the lasing and the tuning cavity, M is the number of active periods in the QCL, τ_{pm} is the photon lifetime for mode m , $T_p(t)$ is the lattice temperature in the tuning cavity, T_0 is the cold finger temperature, m_p and c_p are the mass and specific heat capacity of the laser chip, V_p is the voltage at the tuning cavity and R_{th} is the thermal resistance. The definition and values of all other parameters used in Eqs. (2)–(5) are provided in the Supplementary Information.”

“The RRE model was solved using the fifth-order Runge–Kutta method.²⁰ The optical power at each eigenmode was calculated using the following expression:

$$P_m(t) = \frac{\eta_m h \nu_m S_m(t)}{\tau_{pm}} \quad (6)$$

where η_m is the power out coupling coefficient for mode m .”

“The RRE model was solved to simulate the switch on dynamics in the CC THz QCL. The temporal evolution of a multimode emission to a single mode emission was simulated at tuning cavity current in the range 0–2 A.”

In order to supplement this more detailed description of the simulation process, we have also included the device parameters used to model CC THz QCLs in the Supplementary Information.

To provide a more complete description of the operating principles of the CC THz QCLs we have also included the following texts in section 1 (pages 4–5):

“This way the CC QCL can be described as a single cavity with an effective complex mirror.”

“This is achieved by driving the tuning cavity with wide current pulses, and by varying the amplitude of the current pulses.”

“The emission frequencies, spectral coverage and frequency tuning range realised from such CC lasers is optimised through careful selection of the CC geometry such as the lengths and the ratio of the cavity lengths^{6,16} as well as the tilt of the coupled facets.⁴”

Furthermore, we have also included the following text in the Methods (fabrication) section (pages 13–14), describing the device fabrication:

“QCL ridges with widths of 150 μm and thickness of 11.6 μm were defined using optical photolithography and wet chemical etching using an aqueous etchant solution of H_2SO_4 , H_2O_2 and H_2O , premixed in the ratio 1:8:40. A eutectic alloy of Au/Ge/Ni were deposited using vacuum thermal evaporation to form Ohmic contacts to the QCL. Similarly, cladding layers of Ti/Au were evaporated to form the waveguide. The substrate was thinned to a thickness of ~ 200 μm and a soldering layer of Ti/Au was evaporated on the thinned substrate. A 4.81-mm-long device was cleaved and soldered onto a Cu block using indium. The CC THz QCL was formed after packaging using focused ion beam milling to etch a 12- μm -deep air gap, thereby splitting the cavity into a lasing and a tuning cavity.”

“Finally, the top of the lasing and the tuning cavities were wire-bonded to two separate ceramic pads allowing independent electrical connections to the cavities.”

REVIEWERS' COMMENTS:

Reviewer #1 (Remarks to the Author):

In this revised version, authors have taken into account the various reviewer's questions and remarks. They have also made substantial improvements in the manuscript. As for the weakness of the paper, I think the way authors have reformulated their argumentation is satisfactory albeit I am still not fully convinced that the switch-on dynamics is valid whatever the laser structures. In fact, it seems that both reviewers remain skeptical on this point. To sum, this revised version is now of high quality hence it can deserve for a publication in Nature Communications.

Reviewer #2 (Remarks to the Author):

I have reviewed the revised paper and considered the authors responses. In general, I am satisfied with the responses of the authors and the modifications to the paper. I recommend publication.